# NOX5-induced uncoupling of endothelial NO synthase is a causal mechanism and theragnostic target of an age-related hypertension endotype

Mahmoud H. Elbatreek[1,2]*, Sepideh Sadegh[3], Elisa Anastasi[4], Emre Guney[1,5], Cristian Nogales[1], Tim Kacprowski[3,6], Ahmed A. Hassan[1], Andreas Teubner[7], Po-Hsun Huang[8,9], Chien-Yi Hsu[10,11], Paul M. H. Schiffers[12], Ger M. Janssen[12], Pamela W. M. Kleikers[1], Anil Wipat[4], Jan Baumbach[3,13], Jo G. R. De Mey[1], Harald H. H. W. Schmidt[1]*

1 Department of Pharmacology and Personalised Medicine, School for Mental Health and Neuroscience (MHeNs), Maastricht University, Maastricht, the Netherlands, 2 Department of Pharmacology and Toxicology, School of Pharmacy, Zagazig University, Zagazig, Egypt, 3 Chair of Experimental Bioinformatics, TUM School of Life Sciences Weihenstephan, Technical University of Munich, Munich, Germany, 4 School of Computing, Newcastle University, Newcastle, United Kingdom, 5 Research Programme on Biomedical Informatics, The Hospital del Mar Medical Research Institute and Pompeu Fabra University, Barcelona, Spain, 6 Division of Data Science in Biomedicine, Peter L. Reichertz Institute for Medical Informatics of TU Braunschweig and Hannover Medical School, Brunswick, Germany, 7 Central Animal Facility, CPV, Maastricht University, Maastricht, the Netherlands, 8 Institute of Clinical Medicine, National Yang-Ming University, Taipei, Taiwan, 9 Department of Critical Care Medicine, Taipei Veterans General Hospital, Taipei, Taiwan, 10 Division of Cardiology and Cardiovascular Research Center, Taipei Medical University Hospital, Taipei, Taiwan, 11 Taipei Heart Institute, Division of Cardiology, Department of Internal Medicine, School of Medicine, College of Medicine, Taipei Medical University, Taipei, Taiwan, 12 Department of Pharmacology and Toxicology, Cardiovascular Research Institute Maastricht (CARIM), Maastricht University, Maastricht, the Netherlands, 13 Department of Mathematics and Computer Science, University of Southern Denmark, Odense, Denmark

* melbatreek@ppmlab.net (MHE); hschmidt@ppmlab.net (HHHWS)

**Data Availability Statement:** All relevant data are within the paper and its Supporting Information files. For in silico data, we used the interactome

## Abstract

Hypertension is the most important cause of death and disability in the elderly. In 9 out of 10 cases, the molecular cause, however, is unknown. One mechanistic hypothesis involves impaired endothelium-dependent vasodilation through reactive oxygen species (ROS) formation. Indeed, ROS forming NADPH oxidase (*Nox*) genes associate with hypertension, yet target validation has been negative. We re-investigate this association by molecular network analysis and identify NOX5, not present in rodents, as a sole neighbor to human vasodilatory endothelial nitric oxide (NO) signaling. In hypertensive patients, endothelial microparticles indeed contained higher levels of NOX5—but not NOX1, NOX2, or NOX4—with a bimodal distribution correlating with disease severity. Mechanistically, mice expressing human *Nox5* in endothelial cells developed—upon aging—severe systolic hypertension and impaired endothelium-dependent vasodilation due to uncoupled NO synthase (NOS). We conclude that NOX5-induced uncoupling of endothelial NOS is a causal mechanism and theragnostic target of an age-related hypertension endotype. *Nox5* knock-in (KI) mice represent the first mechanism-based animal model of hypertension.

from IID database which is available at http://ophid.utoronto.ca/iid. For computational data analysis, the MONET tool (https://github.com/BergmannLab/MONET) and SPICi tool (https://compbio.cs.princeton.edu/spici/) were used.

**Funding:** J.B. is grateful for financial support of his VILLUM Young Investigator grant (no. 13154). Financial support to H.H.H.W.S. by the ERC (AdG RadMed '294683' and PoC SAVEBRAIN '737586') and the Horizon 2020 programme (REPO-TRIAL '777111') is gratefully acknowledged. This reflects only the author's view and the European Commission is not responsible for any use that may be made of the information it contains. The funders had no role in study design, data collection and analysis, decision to publish, or preparation of the manuscript.

**Competing interests:** The authors have declared that no competing interests exist.

**Abbreviations:** Ach, acetylcholine; ADMA, asymmetric dimethylarginine; ARRIVE, Animal Research: Reporting of In Vivo Experiments; BMI, body mass index; cGMP, cyclic guanosine monophosphate; DHE, dihydroethidium; DpB, Dierproevenbesluit; DpR, Dierproevenregeling; ecSOD, extracellular superoxide dismutase; eGFR, estimated glomerular filtration rate; ELISA, Enzyme-Linked Immunosorbent Assay; eNOS, endothelial nitric oxide synthase; GCYA1, soluble guanylate cyclase (sGC) alpha; GCYB, sGC beta; GWAS, genome-wide association studies; $H_4Bip$, tetrahydrobiopterin; Hprt, hypoxanthine phospho-ribosyl-transferase; hs-CRP, high-sensitivity C-reactive protein; KGP1, cGMP-dependent protein kinase 1; KI, knock-in; KO, knock-out; L-NAME, L-$N^G$-nitro arginine methyl ester; MAP, mean arterial pressure; NO, nitric oxide; NOS, NO synthase; NOX, NADPH oxidase; PAPA/NO, propylamine propylamine NONOate; PDE5A, phosphodiesterase 5A; PFA, paraformaldehyde; qPCR, quantitative PCR; PPI, protein-protein interaction; ROS, reactive oxygen species; sGC, soluble guanylate cyclase; SHR, spontaneously hypertensive rats; SPD, subnetwork-participation-degree; SPICi, Speed and Performance In Clustering; WoD, Wet op dierproeven; WT, wild type.

## Introduction

Hypertension is of major medical relevance as a risk factor for myocardial infarction, stroke, and other chronic conditions and death [1]. Except for 5% of patients with secondary hypertension (due to renal artery stenosis, adrenal adenomas, pheochromocytomas, and numerous single-gene mutations involving renal transporters [2]), the cause of hypertension remains unknown in the remaining 95% of all cases. In these cases of so-called 'essential hypertension,' treatments have to focus on symptomatic vasodilatory drug therapy and lifestyle management. Even this symptomatic antihypertensive therapy is sometimes ineffective, i.e., in treatment-resistant hypertension, and requires a high number to treat, with many patients still experiencing adverse outcomes such as stroke and myocardial infarction [3].

One molecular mechanism of hypertension that has been suggested for decades is oxidative stress, i.e., an unphysiological production of reactive oxygen species (ROS), which in blood vessels interferes with vasodilation by the endothelium-derived relaxing factor, nitric oxide (NO) [4]. No hypertension-relevant cellular source of ROS, however, has been identified to either prove this hypothesis or exploit it for a mechanism-based or even curative clinical therapy.

Recent genome-wide association studies (GWAS) [5] in search of hypertension risk genes point towards NADPH oxidases (*Nox*), the only known enzyme family dedicated to ROS formation, in particular the genes *Nox4* and *Nox5*. This matches preclinical studies, excluding other vascular NOX isoforms, i.e., NOX1 and NOX2, for causing hypertension, unless animals are infused with pro-hypertensive agents [6–8]. Mice overexpressing p22phox (an essential component of NOX 1–4) or lacking the antioxidant extracellular superoxide dismutase (ecSOD) in the vasculature develop hypertension upon ageing [9]. Thus, the data regarding the role of NOX1 and NOX2 regarding preclinical hypertension are controversial and suggest that these isoforms may contribute to basal blood pressure regulation but not hypertension [6, 8, 10, 11]. With respect to NOX4, this isoform is widely expressed, but it appears irrelevant for blood pressure or hypertension [12] and is rather vasoprotective [13]. Regarding NOX5, this enzyme is physiologically expressed in vascular endothelial cells of human blood vessels and may be associated with diabetic nephropathy [14–16]; mice expressing human *Nox5* in vascular smooth muscle cells are, however, normotensive [17]. Thus, the role of NOX5 in hypertension is unclear.

Network medicine [18] predicts that, for most particularly complex diseases, not a single protein but protein modules, i.e., sub-graphs of the interactome, are in fact relevant [19–21]. Therefore, we set out to re-investigate the association of NOX with hypertension and NO-dependent vasodilation using 3 complementing unbiased in silico approaches and to validate any prediction both in mice and—if possible—also human patient samples.

## Results

### NOX5 is the only direct neighbor of endothelial NO-cyclic guanosine monophosphate (cGMP) signaling

To explore the possible link between NOX isoforms and hypertension and NO-dependent vasodilation, we constructed a pruned molecular subnetwork from the first neighbors of NOX family members and NO-cGMP-related proteins as seed nodes in the experimentally validated interactome obtained from the IID [22] interactome database. These included NOX1, NOX4, NOX5, NO synthase 1 (NOS1), NOS3, soluble guanylate cyclase (sGC) alpha-1 (GCYA1), sGC alpha-2 (GCYA2), sGC beta-1 (GCYB1), phosphodiesterase 5A (PDE5A), PDE9A, and cGMP-dependent protein kinase 1 (KGP1), but not NOX2 and NOS2. The resulting

subnetwork was further pruned according to the subnetwork-participation-degree (SPD) to correct for hub nodes (i.e., proteins that occurred mainly because of their high number of interactions in the whole network). This resulted in a disease module consisting of several connected components, which revealed that all NOX isoforms but NOX5 were excluded as a close neighbor of endothelial NO-cGMP signaling. NOX5 fell into the same connected component with the subunits of the NO receptor, GCYA1, GCYA2, and GCYB1, and with endothelial NOS (NOS3) (Fig 1A). From IID, this connection is based on a physical interaction suggested by high-throughput affinity chromatography [23].

To cross-check our in silico findings of a NOX disease module, we employed 2 additional computational network module identification methods, global modularity optimization and agglomerative local search, both of which have been top performers in the recent Module Identification DREAM Challenge [24]. In brief, the global modularity optimization approach combines multiple module detection algorithms to avoid suboptimal partitions resulting from individual algorithms [25]. The agglomerative local method uses the Speed and Performance In Clustering (SPICi) algorithm [26] to optimize the local density of modules around seed nodes. With all 3 in silico methods, we reached the same conclusion: exclusion of all NOXs, except NOX5, as a direct neighbor of endothelial NO-cGMP signaling (Fig 1A).

## NOX5 protein levels are increased in hypertensive patients

To test this causal endothelial NOX5 hypothesis for human hypertension, we enrolled consecutive outpatients with essential hypertension and a baseline estimated glomerular filtration rate (eGFR) $\geq$30 mL/min/1.73 m$^2$. Study participants were divided into 3 groups, healthy ($n$ = 10), hypertensive patients with normoalbuminuria ($n$ = 20), and hypertensive patients with moderately increased albuminuria (previously termed microalbuminuria) ($n$ = 20). The baseline characteristics of the patients are listed in Table 1. To measure NOX5 protein levels, circulating endothelial microparticles, i.e., membrane vesicles that are released from endothelial cells upon cellular activation or cell death and that carry endothelial proteins [27], were isolated from plasma of the participants (Fig 1B). We observed higher NOX5 protein levels in endothelial microparticles of hypertensive compared with normotensive individuals, and within hypertensive individuals, patients with microalbuminuria showed even higher NOX5 protein levels (Fig 1C). These data suggest that NOX5 levels are associated with hypertension and correlate with disease severity. Hypertension is rather an umbrella term that may involve different molecular mechanisms all resulting in a similar phenotype, i.e., elevated blood pressure. NOX5-dependent hypertension may be such an endotype but apply only to a subset of patients [28–30]. We therefore performed a subgroup analysis of all hypertensive patients, and indeed, NOX5 levels showed a bimodal distribution (Fig 1D). Based on this, approximately every fourth hypertensive patient would fall into a high NOX5 mechanotype, which according to the protein-protein interactions (PPIs) would cause NO-cGMP signaling dysfunction. With respect to other vascular NOX isoforms, i.e., NOX1, NOX2, and NOX4, their levels did not differ significantly between normotensive and hypertensive individuals. However, patients with microalbuminuria showed higher NOX1, but lower NOX2, levels compared to normotensive individuals (S1 Fig). The higher NOX1 levels in those patients is in agreement with previous preclinical data showing that NOX1 is not associated with elevation of blood pressure but with renal oxidative stress in a model of chronic hypertension [6] and renal pathology [31–33].

In addition, we determined plasma asymmetric dimethylarginine (ADMA) levels, a biomarker of NOS uncoupling and endothelial dysfunction [34]. We found that ADMA levels were significantly increased in hypertensive patients compared to healthy individuals (S1 Fig), which is in agreement with previous findings [35, 36].

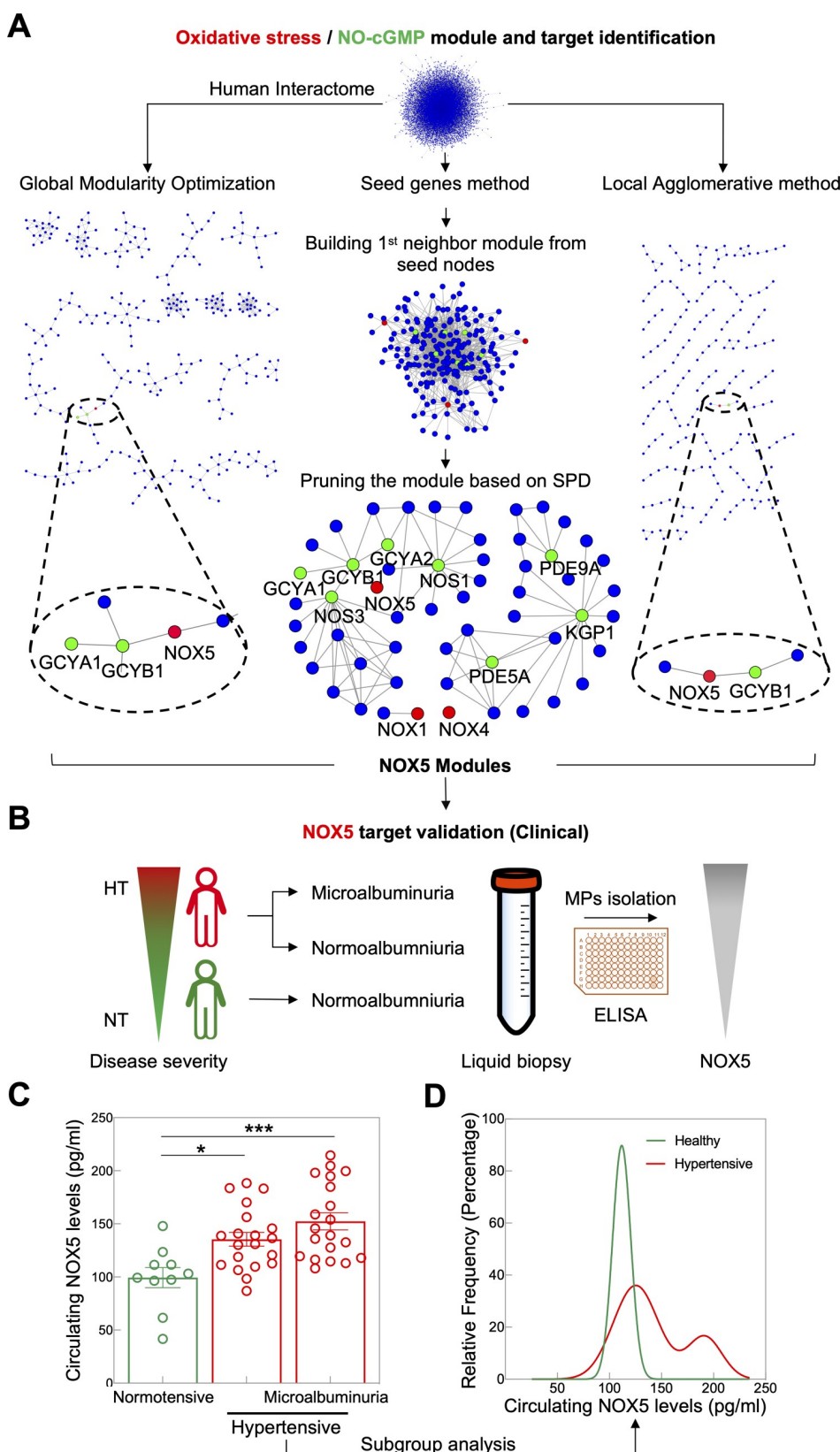

**Fig 1. Identification of NOX5 as direct neighbor of endothelial NO-cGMP signaling and clinical validation in hypertension.** (A) NOX module was constructed by first neighbor subnetwork pruned based on SPD (the middle panel) where NOX isoforms (red nodes) and NO-cGMP related proteins (green nodes) were used as seed nodes. The resulting NOX module was confirmed with 2 disease module identification methods, global modularity optimization (the left panel) and the agglomerative local method (the right panel). All the methods identified NOX5 as the closest link to NO-cGMP signaling and excluded NOX1–4. (B) NOX5 levels in endothelial microparticles (MPs) isolated from plasma of NT, normoalbuminuric individuals and HT normoalbuminuric and microalbuminuric patients were measured by ELISA. (C) NOX5 levels were increased in HT patients with normoalbuminuria ($n = 20$) compared to NT individuals ($n = 10$). NOX5 levels were even higher in HT patients with microalbuminuria ($n = 20$). Comparison between groups was done by one-way ANOVA followed by Tukey's multiple comparisons test. (D) Subgroup analysis of all HT patients shows a bimodal distribution ($p = 0.0007$, two-tailed F-test, adjusted $r^2 = 0.9973$), where only the right but not the left subgroup peak of HT patients has NOX5 levels that are significantly different from the healthy individuals. All data are represented as mean ± SEM of $n$ independent experiments $*p = 0.014$, $***p = 0.0003$. All raw data are included in the S1 Data file. cGMP, cyclic GMP; ELISA, enzyme-linked immunosorbent assay; GCYA, soluble guanylate cyclase alpha; GCYB, soluble guanylate cyclase beta; HT, hypertensive; KGP1, cGMP-dependent protein kinase 1; MP, microparticle; NO, nitric oxide; NOS, NO synthase; NOX, NADPH oxidase; NT, normotensive; PDE, phosphodiesterase; SPD, subnetwork-participation-degree.

**Table 1. Baseline characteristics in healthy individuals and hypertensive patients.**

| | Healthy individuals ($n = 10$) | HTN patients without albuminuria ($n = 20$) | HTN patients with microalbuminuria ($n = 20$) | *p*-Value |
|---|---|---|---|---|
| **Age (y)** | 44 ± 4 | 56 ± 3 | 60 ± 3 | 0.012 |
| **Men** | 7 (70%) | 12 (60%) | 13 (65%) | 0.859 |
| **Diabetes** | 0 (0%) | 3 (15%) | 6 (30%) | 0.118 |
| **BMI** | 24.1 ± 0.94 | 26.1 ± 0.76 | 26.9 ± 0.73 | 0.096 |
| **Smoking** | 0 (0%) | 7 (35%) | 9 (45%) | 0.042 |
| **T. Chol** | 178 ± 11 | 193 ± 9 | 199 ± 8 | 0.358 |
| **Triglyceride** | 158 ± 42 | 146 ± 19 | 214 ± 27 | 0.139 |
| **HDL** | 50 ± 3 | 47 ± 3 | 41 ± 2 | 0.081 |
| **LDL** | 97 ± 11 | 117 ± 8 | 115 ± 9 | 0.360 |
| **Fasting glucose** | 96 ± 6 | 104 ± 8 | 120 ± 7 | 0.107 |
| **Serum Cr** | 0.81 ± 0.07 | 0.91 ± 0.05 | 1.01 ± 0.07 | 0.144 |
| **Uric acid** | 6.2 ± 0.53 | 6.1 ± 0.31 | 6.5 ± 0.29 | 0.656 |
| **GFR** | 87.0 ± 5.69 | 83.2 ± 3.57 | 79.3 ± 5.52 | 0.606 |
| **FRS** | 5.2 ± 1.73 | 8.2 ± 1.78 | 10.9 ± 1.9 | 0.168 |
| **ACR** | 0.008 ± 0.0009 | 0.010 ± 0.0001 | 0.059 ± 0.0008 | < 0.001 |
| **hs-CRP** | 0.56 ± 0.33 | 0.25 ± 0.03 | 0.52 ± 0.09 | 0.188 |
| **Adiponectin** | 16.0 ± 2.75 | 20.4 ± 3.08 | 18.5 ± 2.99 | 0.671 |
| **NT-pro-BNP** | 75.1 ± 11.95 | 80.9 ± 8.76 | 95.8 ± 16.9 | 0.572 |
| **Medications** | | | | |
| ACE-I | 0 (0%) | 4 (20%) | 1 (5%) | 0.143 |
| ARB | 0 (0%) | 14 (70%) | 16 (80%) | < 0.001 |
| CCB | 0 (0%) | 13 (65%) | 14 (70%) | < 0.001 |
| Beta-blocker | 0 (0%) | 2 (10%) | 9 (45%) | 0.005 |
| Thiazides | 0 (0%) | 6 (30%) | 8 (40%) | 0.069 |
| Statin | 1 (10%) | 3 (15%) | 7 (35%) | 0.185 |

Values are mean ± SEM or number (%).

**Abbreviations:** ACE-I, angiotensin-converting enzyme inhibitor; ACR, albumin/creatinine ratio; ARB, angiotensin II receptor blocker; BMI, body mass index; CCB, calcium channel blocker; Cr, creatinine (mg/dL); FRS, Framingham risk score (%); GFR, glomerular filtration rate (mL/min/1.73 m$^2$/year); HDL, high-density lipoprotein (mg/dL); hs-CRP, high-sensitivity C-reactive protein (mg/dL); HTN, hypertension; LDL, low-density lipoprotein (mg/dL); T. Chol, total cholesterol (mg/dL); T-pro-BNP, N terminal pro-brain natriuretic peptide (pg/mL)

## Endothelial NOX5 induces age-related hypertension

In the absence of NOX5 specific inhibitors, we tested the possible role of NOX5 in endothelial NO-cGMP signaling dysfunction and hypertension in mice. Mice, however, lack the *Nox5* gene. We therefore analyzed a knock-in (KI) mouse model expressing human *Nox5* in its physiological endothelial cell location [37] (Fig 2A). In young (9–15 weeks old) NOX5 KI mice of both genders (*n* = 19–20), systolic blood pressure, diastolic blood pressure, and mean arterial

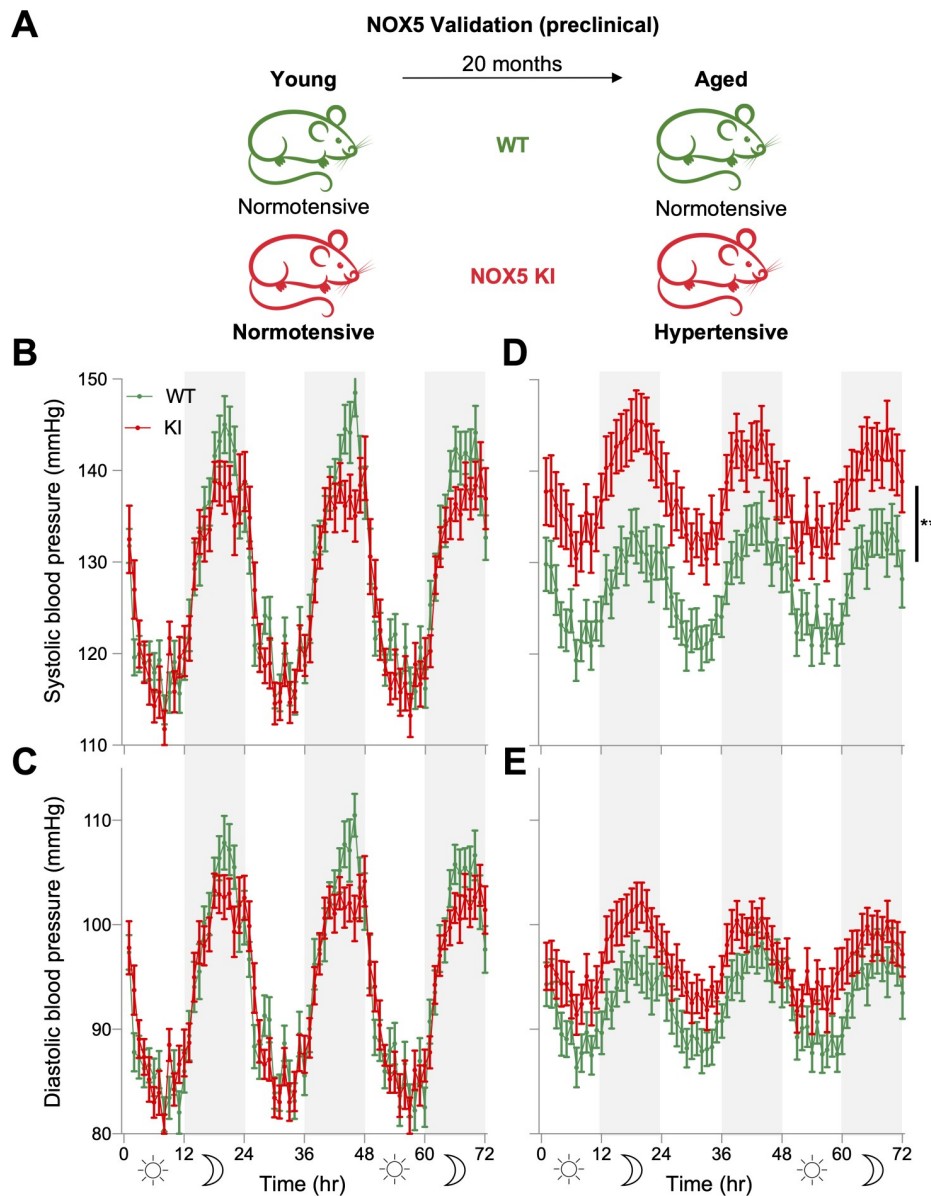

**Fig 2. Preclinical validation of NOX5 in hypertension.** (A) Both young WT and NOX5 KI mice were normotensive; however, only KI mice become hypertensive upon ageing. (B, C) There was no significant difference in systolic (B) and diastolic (C) blood pressure between young WT (*n* = 19) and KI (*n* = 20). (D, E) Aged KI mice (*n* = 33) had higher systolic (D) but similar diastolic (E) blood pressure compared to WT (*n* = 31). Telemetry data were analyzed by two-way repeated measures ANOVA followed by Sidak's multiple comparisons test. All data are represented as mean ± SEM of *n* individual animals **$p < 0.01$. All raw data are included in the S1 Data file. KI, knock-in; NOX5, NADPH oxidase 5; WT, wild type.

pressure (MAP) were, however, not different from age- and sex-matched wild-type (WT) mice (Fig 2B and 2C and S2 Fig). Upon aging (68–87 weeks), though, systolic blood pressure and MAP were significantly elevated throughout the day in KI ($n$ = 33) compared to age- and sex-matched WT mice ($n$ = 31) (Fig 2D and S2 Fig). Diastolic blood pressure remained unmodified (Fig 2E) as did heart to body weight ratio (S3 Fig) indicating that there was no cardiac hypertrophy in KI mice, which is in line with the late (age-dependent) development of hypertension. In addition, cardiac hypertrophy does not necessarily exist with hypertension. For example, other hypertensive animal models such as endothelial NOS (eNOS) knock-out (KO) mice do not have cardiac hypertrophy [38, 39]. This is also in agreement with clinical observations that not all hypertensive patients (subgroups) have cardiac hypertrophy [40–42]. Notably, there was no difference in blood pressure between aged KI male (72-hour MAP, 120 ± 3.4, $n$ = 15, $p$ = 0.13) and female (114 ± 2.1, $n$ = 18) mice.

Taken together, our observations indicate that in mice, expression of NOX5 in the endothelium leads, upon aging, to a selective elevation of systolic arterial blood pressure. Having established the potential of NOX5 to induce a hypertensive phenotype, we wanted to test the mechanistic link to vascular NO-cGMP signaling as suggested from the in silico network analysis.

## Endothelial NOX5 causes endothelial dysfunction by uncoupling NOS

In the thoracic aorta, femoral artery, and saphenous artery isolated from aged KI and WT mice of both genders ($n$ = 9), we analyzed the structural, smooth muscle, and endothelial vasomotor properties. Collectively, these blood vessels cover the entire range of large elastic conduit, muscular conduit, and small muscular resistance-sized arteries, respectively. In the thoracic aorta, femoral artery, and saphenous artery of the aged animals, the relation between resting wall tension and arterial lumen diameter did not differ between KI and WT mice (S4 Fig). It is therefore unlikely that the blood pressure phenotype of the KI mice resulted from stiffening or inward remodeling of the conduit or resistance arteries. This is in agreement with previous studies that structural stiffening of arteries is not a general observation in rodent models of (essential) hypertension [43–46]. Also, clinical data, especially from aged hypertensive patients, are in line with this finding [47–50].

To test the effect of endothelial NOX5 on endothelium-dependent, NO-cGMP–mediated relaxation, arterial segments were pre-contracted by either depolarization ($K^+$), $\alpha_1$-adrenergic activation (phenylephrine), or endothelin-1; vasorelaxation was then induced by acetylcholine (Ach) the classical endothelium-derived relaxing factor stimulant [51]. In femoral arteries, irrespective of whether pre-contracted with $K^+$, phenylephrine, or endothelin-1, the amplitudes of Ach-induced relaxing responses were significantly smaller in KI compared to WT mice (Fig 3A) (S5 Fig). Conversely, in the saphenous artery (Fig 3B) (S5 Fig) and thoracic aorta (S5 Fig), Ach-induced relaxing responses did not differ between KI and WT mice. When comparing ours to previous studies, the Ach-induced relaxation of the saphenous arteries of both mice groups seemed attenuated [52–54]. There are 2 possible explanations for this discrepancy. First, we have used very old mice and other studies used young mice [52–54]. Second, we have used mice with a mixed genetic background (80% 129/Sv and 20% C57Bl6), and previous data showed that endothelium-dependent relaxation is smaller in 129/Sv compared with C57Bl6 mice [55].

In segments of the thoracic aorta and femoral artery contracted with 256 nM endothelin-1, the relaxing effects of Ach were reversed by 100 μM L-N$^G$-nitro arginine methyl ester (L-NAME) (a pharmacological inhibitor of NOSs), and this did not differ significantly between preparations of KI and WT mice (S5 Fig). In the saphenous arteries of both types of mice,

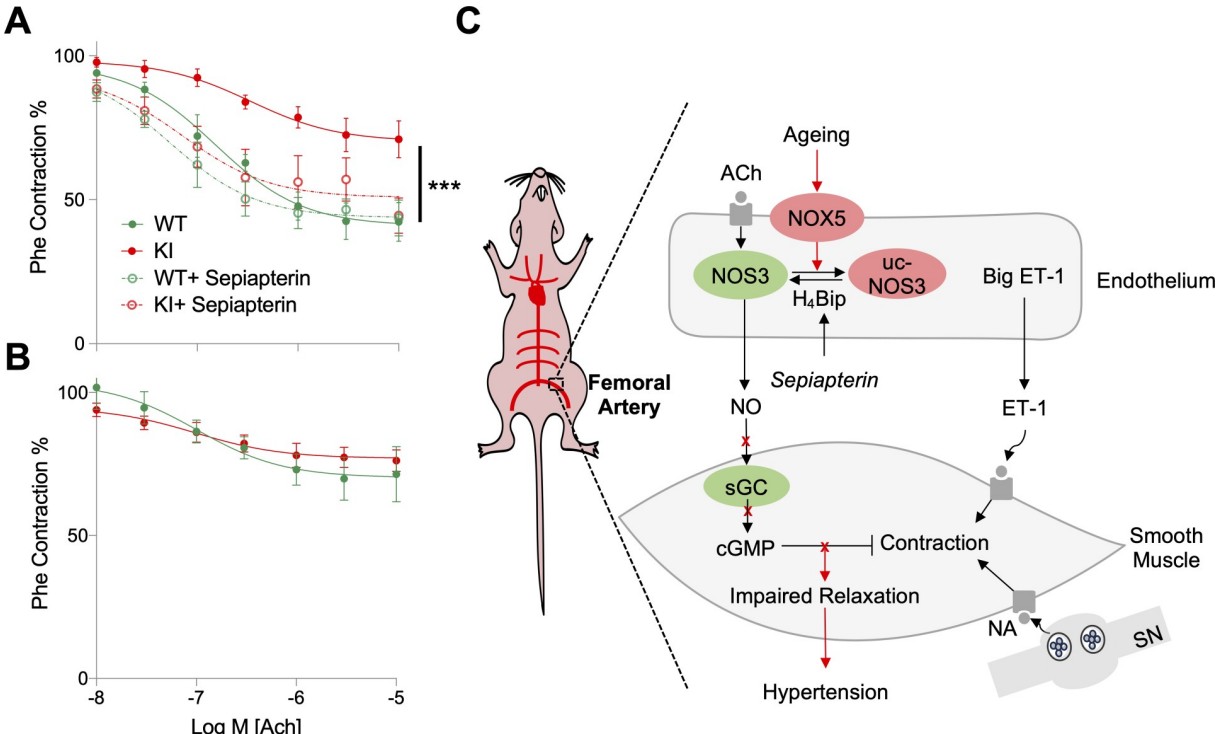

**Fig 3. Endothelial NOX5 induces endothelial dysfunction and hypertension by uncoupling NOS.** (A) Femoral arteries of aged KI mice ($n =$ 9) precontracted with Phe had less responsiveness to Ach-induced relaxation compared to WT ($n = 9$), while pretreatment with sepiapterin (100 μM) improved relaxation in KI ($n = 4$), which was not different from WT ($n = 3$). (B) Saphenous arteries of aged KI mice ($n = 9$) precontracted with Phe showed no difference in Ach-induced relaxation compared to WT ($n = 8$). Myograph data were analyzed by two-way ANOVA followed by Sidak's multiple comparisons test. (C) Schematic representation of NOX5-induced age-dependent hypertension. In ageing, endothelial NOX5 is activated and interferes with normal NO-cGMP signaling, which results in impaired vascular smooth muscle relaxations and raised blood pressure. All data are represented as mean ± SEM of $n$ individual animals ***$p < 0.001$. All raw data are included in the S1 Data file. Ach, acetylcholine; cGMP, cyclic GMP; ET-1, endothelin-1; H$_4$Bip, tetrahydrobiopterin; KI, knock-in; NA, noradrenaline; NO, nitric oxide; NOS3, endothelial nitric oxide synthase; NOX5, NADPH oxidase; Phe, phenylephrine; sGC, soluble guanylate cyclase; SN, sympathetic nerve; uc-NOS3, uncoupled NOS3; WT, wild type.

L-NAME did not modify the relaxing effects of Ach (S5 Fig). This is in line with our earlier findings indicating that in these resistance-sized arteries, Ach-induced relaxation is mediated by endothelium-dependent hyperpolarization but not NO [52–54].

To check whether this caliber-specific effect on vasomotor function was due to differential expression of NOX5 along the systemic arterial tree, we measured *Nox5* gene expression by quantitative PCR (qPCR). There was, however, no difference in *Nox5* gene expression between thoracic aorta, femoral arteries, and saphenous arteries of *Nox5* KI mice (S6 Fig). These data suggest an aging-dependent dysfunction of endothelial NO-cGMP signaling due to NOX5 and that this effect is not uniformly distributed along the systemic arterial tree.

We next tested whether, alternatively, chronic changes in the underlying arterial smooth muscle layer could have contributed to the observed blood pressure and vasomotor pheno-types in aged *Nox5* KI mice. Contractile responses to K$^+$ and the sensitivity and maximal responsiveness to phenylephrine and endothelin-1 did not, however, differ between KI and WT mice in all arterial segments (S7 Fig). Also, the blunting of agonist-induced contractile responses by indomethacin was similar in the thoracic aorta of the KI and WT mice (S7 Fig).

To test which component of NO-cGMP signaling was most likely affected, we tested an uncoupling effect on endothelial NOS [56] or oxidative damage of the NO receptor, soluble

guanylate cyclase (sGC) yielding oxidized or heme-free apo-sGC [57, 58]. To test for sGC/apo-sGC, relaxing responses to the NO donor compound and sGC stimulator, propylamine propylamine NONOate (PAPA/NO) (0.01–10 µM) [59], and to the apo-sGC activator, BAY 60–2770 (0.01–10 µM) [57], were analyzed. Neither the PAPA/NO (S8 Fig) nor the BAY 60–2770 response (S8 Fig) differed between WT and KI mice. These observations suggested that sGC was not dysfunctional in aged NOX5 KI mice.

Uncoupling of endothelial NOS is considered a major cause of endothelial dysfunction characterized by decreased NO formation and increased superoxide production and occurs mainly when ROS oxidize the NOS cofactor tetrahydrobiopterin ($H_4$Bip) [60]. When we incubated femoral arteries of aged NOX5 KI mice, precontracted with phenylephrine, with the $H_4$Bip precursor sepiapterin (100 µM), Ach-induced relaxations were greatly improved and became indistinguishable from those in WT (Fig 3A). Also, femoral arteries of aged NOX5 KI mice showed higher superoxide formation (as indicated by dihydroethidium [DHE] staining) than WT, and this increase was inhibited by pretreatment with the NOS inhibitor, L-NAME (S9 Fig). We also tested whether the impaired Ach-induced relaxation in the femoral arteries of *Nox5* KI mice could be reversed by adding antioxidants to the bath medium. Neither 10 µM N-acetylcysteine nor 100 µM tempol, however, were effective (S10 Fig). Collectively, these data suggest that endothelial NOX5 induces endothelial dysfunction by uncoupling of endothelial NOS leading to impaired endothelium-dependent relaxation of muscular conduit arteries and thus systolic hypertension (Fig 3C).

## Discussion

Based on human genetic, human clinical, and genetic preclinical mechanistic validation, we here report the first identified causal molecular mechanism of human systolic hypertension associated with aging. This endotype affects approximately one in 4 patients and molecularly consists of a NOX5-induced uncoupling of endothelial NOS followed by impaired endothelium-dependent vasodilation in muscular conduit arteries. Detection of elevated levels of NOX5 in circulating microparticles appears to serve as a mechanism-based liquid biopsy marker to stratify patients for therapeutic intervention. Based on our in vivo validation, such an intervention may include the $H_4$Bpi precursor and NOS recoupling agent sepiapterin and (once such compounds should become clinically available) [61–64] a NOX5 inhibitor.

Here, we used a KI mouse model that expresses human NOX5 in endothelial cells and white blood cells mimicking, to a large extent, the physiological pattern of NOX5 expression in humans [37]. While we cannot exclude a potential contribution of leukocytic NOX5 on hypertension in aged animals in our in vivo experiments, a contribution of adhering leukocytes is unlikely in our ex vivo experiments. Expression of human *Nox5* in mice led with aging to severe systolic hypertension. This was not due to stiffening, structural remodeling, or increased sensitivity to vasoconstrictor stimuli in the systemic arterial tree. It rather resulted from a regionally selective and specific attenuation of NO-mediated endothelium-dependent relaxation in medium-sized muscular conduit arteries via uncoupling of NOS.

Collectively, our data warrant clinical proof-of-concept trials aiming at a theragnostic [65] strategy in carefully stratified hypertensive patients based on the detection of elevated NOX5, ROS overproduction, e.g., by oxidatively modified proteins, and mechanism-based functional repair by a network pharmacology approach that inhibits NOX5 and recouples NOS. If successful, this will be the first case of a molecular redefinition of the phenotypic disease definition, essential hypertension, and the first step towards precision medicine in a currently high number-needed-to-treat indication. This may be the case for about a fourth of all hypertensives. Our estimate, though significant, is limited by the relatively small number of patients in

the present study and needs to be validated in larger and international hypertension cohort. In addition, other factors that might affect NOX5 and endothelial function, including smoking, comorbidities, and antihypertensive drugs, warrant investigation in future studies.

Because NOX5 is also involved in a worse outcome in stroke [37]—for which hypertension is a major risk factor [66]—and correlates with atherosclerosis [67], this approach may lower not only blood pressure but also 2 major consequences of hypertension, stroke and myocardial infarction.

On a broader scale, our present 3-fold interactome-based approach for disease module discovery, preclinical and clinal validation appears to be applicable to a wide range of common or complex diseases. We here identify NOX5 as the missing link between ROS and impaired NO signaling. The full module consists of NOX5, NOS3, the different subunits of the NO receptor sGC, and the phosphodiesterases PDE5 and PDE9, as well as the cGMP-dependent protein kinase, PKG1. The dysregulation of such a module may be best treated by multiple drugs targeting different protein components. In the present case, the WT mice mimic pharmacological NOX5 inhibition (not yet available) and the treatment with sepiapterin, NOS recoupling.

Disease module construction is a young research field at the interface of biomedicine and bioinformatics. We began with a seed gene-based approach, based on clinically validated proteins. *Noxs* were suggested by GWAS in old [5] but not in younger patients [68] and are the only known enzyme family solely dedicated to ROS formation [69]. The NO-cGMP pathway is important for blood pressure regulation and its dysfunction a hallmark of hypertension [70]. Our approach yielded a module containing a sufficient set of NO-cGMP signaling components, and, importantly, NOX5 was the sole ROS source. We confirmed our findings independently by 2 complementing in silico network module detection approaches. As a potential limitation of our in silico findings, the interaction between NOX5 and cGMP-related proteins has been shown experimentally so far in only one study [23].

Our clinical data are yet still correlative and deserve extensive further investigation. Endothelial microparticles are, however, well established surrogate biomarkers associated with hypertension and its progression [71, 72]. They produce ROS, contain NOX, induce endothelial dysfunction, and impair endothelium-dependent relaxation [73]. In human endothelial cells, angiotensin II—the target of clinically used angiotensin type 1 receptor blockers and angiotensin-converting enzyme inhibitors—and the pro-hypertensive autacoid endothelin-1 increase NOX5 expression (gene and protein) and activity [74]. In addition to its physiological vascular expression in endothelial cells, NOX5 is also increased (gene, protein, and activity) to higher levels in human renal proximal tubule cells of hypertensive patients [75], which may contribute to the observed correlation of NOX5, blood pressure, and microalbuminuria. Induction of NOX5 in smooth muscle cells does not cause hypertension per se but correlates with advanced atherosclerotic lesions, and diseased coronary arteries show high NOX5 expression and activity [67]. Hypertension is rather an umbrella term for different blood pressure–elevating mechanisms. Some of these may be related to worse clinical outcomes, some not. NOX5-dependent hypertension, however, appears to be disease relevant as this molecular mechanism also leads to or aggravates hypertension-associated clinical outcomes, stroke [37], myocardial infarction [76], and renal failure [75, 77]. Together, the human NOX5 data provide an exciting clinical lead, which we are now planning on following up in a major multicenter, multinational clinical trial (HYPERNET).

GWAS had delivered 2 *Nox* candidate genes, *Nox4* and *Nox5*. Knocking out *Nox4* is without a blood pressure phenotype [12, 78]; in many models, *Nox4* was rather a vasoprotective [13, 79, 80]. The enzymatic product of NOX4, $H_2O_2$, activates NOS [81] and is an alternative endothelium-derived relaxing factor in its own right [82–84]. This left *Nox5* as a candidate gene, which was confirmed by first neighbor analysis linked to NO-cGMP signaling. Unlike

NOX4, NOX5 produces superoxide, which can either uncouple endothelial NOS in hypertension by oxidation of $H_4Bip$ [85, 86] in a sepiapterin-reversible manner [87] or scavenge NO to form peroxynitrite, which may also act as a cytotoxic agent or vasodilator [88, 89]. The mechanistic validation of this role of NOX5 in hypertension was performed in a preclinical mouse KI model expressing *Nox5*, not present in the mouse genome, in the physiological cell type, endothelial cells where NOX5 is important for endothelial migration and angiogenesis [90]. In mice expressing *Nox5* non-physiologically in smooth muscle cells, blood pressure is normal, and angiotensin II–induced high blood pressure is not augmented [17]. Of note, in other animal models of hypertension—e.g., spontaneously hypertensive (SHR) and Dahl salt-sensitive rats, in which NOX5 is not present—antioxidants can reduce blood pressure [91–93]. This may suggest that, if relevant also to humans, other endotypes of hypertension may involve other ROS sources (mitochondria, xanthine oxidase, or myeloperoxidase).

Of particular interest is the regionally selective effect of NOX5, which was reminiscent of the vascular heterogeneity in age-related endothelial dysfunction [94, 95]. It was present in muscular femoral conduit arteries but not in the small resistance-sized saphenous arteries (lumen diameter <250 μm) of the KI mice and could thus selectively result in systolic hypertension with accompanying elevation of arterial pulse pressure. However, we have not considered other small resistance arteries, including renal arteries, which control sodium handling and blood pressure and are more relevant in secondary renovascular hypertension [96]. This effect of NOX5 was not reversed in our ex vivo experiments by antioxidants, presumably because NOS uncoupling was already established chronically in vivo by NOX5-derived superoxide or NO production was reduced as part of age-related endothelial dysfunction. Preclinically, sepiapterin or $H_4Bip$ lower elevated blood pressure induced by NOS uncoupling [97, 98]. As a potential limitation to our study, we did not re-*examine in vivo* in *Nox5* KI mice the efficacy of sepiapterin to reverse NOS uncoupling and hypertension. Another limitation and general pitfall of all animal models (particularly the KI mouse model) are differences between human and mouse physiology and that expressing a human protein in mice may not wholly reflect its role in human pathology [99].

With respect to the clinical dimension of our findings, our observation that plasma ADMA levels were elevated in hypertensive patients speaks in favor of NOS uncoupling [35, 36]; yet a similar mechanistic link to NOX5 as we obtained pre-clinically in *Nox5* KI mice would require the ex vivo analysis of isolated human blood vessels or an interventional trial with sepiapterin. Indeed, sepiapterin analogs, i.e., folic acid and $H_4Bip$, are clinically effective to reduce elevated blood pressure by improving endothelial function [100, 101]. Folic acid, either alone [102] or in combination with antihypertensives [103, 104], reduces the risk of cardiovascular and cerebrovascular events in hypertensive patients.

Overall, our findings using in silico network approaches as well as further clinical and preclinical validation explain the long-observed correlation between oxidative stress, endothelial dysfunction, and systolic hypertension. Humanized endothelial cell *Nox5* KI mice represent the first mechanism-based animal model of human age-related hypertension and endothelial dysfunction. Therapeutically, NOX5 inhibition and NOS recoupling, ideally in combination with mechanistic biomarker stratification, e.g., based on endothelial microparticle liquid biopsies, represents a first-in-class mechanism-based approach for curative antihypertensive therapy obviating the need for symptomatic vasodilators.

## Materials and methods

### Ethics statement

This human subjects study was approved by the research ethics committee of Taipei Veterans General Hospital (Taipei Veterans General Hospital Institutional Review Board, No.: 96-12-

42A), and all participants provided their written informed consent. The study complied with the Declaration of Helsinki. It was approved by the appropriate health authorities, an independent ethics committees, and the Institutional Review Board of Taipei Veterans General Hospital.

All animal experimental research was approved by the Animal Welfare Body (IvD) of Maastricht University, the Animal Ethics Committee (Dier Experimenten Commissie [DEC]), and the Central Authority for Scientific Procedures on Animals (Centrale Commissie Dierproeven [CCD]). The DEC project numbers are 2012–123 and PV 2017–029, and the license number is AVD1070020174486. The animal care and use protocol adhere to the ARRIVE guidelines ("Animal Research: Reporting of In Vivo Experiments"; https://arriveguidelines.org/), Wet op dierproeven (WoD; 2014, https://wetten.overheid.nl/BWBR0003081/2019-01-01), the Dierproevenregeling (DpR; 2014, https://zoek.officielebekendmakingen.nl/stcrt-2014-34746.html), the Dierproevenbesluit (DpB; 2014, https://wetten.overheid.nl/BWBR0035866/2014-12-18), and the EU Directive 2010/63 (2010, https://eur-lex.europa.eu/eli/dir/2010/63/2019-06-26).

## Study design

The human subjects sample size was determined by G*Power software. For mice, we used a power analysis according to the formula $n = 2 \times s2 \times (Za/2 + Zb)2/D2$ (L. Sachs, Angewandte Statistik, Springer, 1983, Berlijn, Springer Verlag). Human subjects with history or clinical evidence of angina, myocardial infarction, congestive heart failure, peripheral vascular disease, inflammatory disease, or any disease predisposing to vasculitis were excluded. Causes of secondary hypertension were excluded by appropriate investigations. Patients with stage 4 and 5 chronic kidney disease (GFR $<$ 30 mL/min/1.73 m$^2$) were also excluded. Human samples were allocated to different groups based on blood pressure and albuminuria values. Mice were allocated to experimental groups according to genotypes. Investigators were blinded to the experimental groups. Replicate experiments were successful. All experiments were reproduced at least 3 times with independent biological samples.

## In silico methods

We extracted a molecular subnetwork from experimentally validated PPIs from the IID [22] database (interactome) version 2018–11 using NOX family members and NO-cGMP related proteins as seed nodes. This set of seeds comprises NOX1, NOX4, NOX5, NOS1, NOS3, GCYA1, GCYA2, GCYB1, PDE5A, PDE9A, and KGP1. We obtained the subnetwork induced by all first neighbors of the seed genes from the interactome. The induced subnetwork was then pruned according to the SPD, defined as the degree of the node (protein) inside the subnetwork normalized by the degree of the node in the full interactome. The SPD quantifies how enriched the interactions of a given protein are in a given subnetwork. This way, we emerged a weighted disease module, which is represented by a set of connected components and some single nodes. We selected an SPD cutoff corresponding to 80% of the cumulative sum of the percentage of the nodes as a cutoff value in the pruning step, as this includes most module-specific interactions while excluding most nonspecific nodes. The final subnetwork consisted of 56 proteins and 83 PPIs.

Moreover, we applied the 2 top-ranked disease network module identification methods from the Module Identification DREAM Challenge [24] to the interactome. We selected these methods from 2 complementary categories of methods, global and local. The main difference between these 2 methods is that global methods exploit the global structure information of PPIs networks, while the local method considers only local neighbor information. The global modularity optimization method (M1) of the DREAM challenge bundled in the MONET tool

and the agglomerative local method (L1) from SPICi tool [26] have been selected (as best-performers in their categories) and applied to find disease modules in the interactome. Note that M1 is an ensemble approach combining multiple module detection algorithms to avoid suboptimal partitions resulting from individual algorithms [25], which notably works without any seed nodes in this tool. The agglomerative L1 method clusters the network greedily, starting from automatically selected local seeds with a high weighted degree. This algorithm improves the local density of modules in the neighboring region of seed nodes.

## Human study participants

We designed the current study based on a previous study in which we had enrolled consecutive outpatients with essential hypertension and a baseline eGFR $\geq$ 30 mL/min/1.73 m$^2$ at Taipei Veterans General Hospital between April 2008 and December 2008 [105, 106]. Hypertension was defined as systolic blood pressure $\geq$140 mmHg, diastolic blood pressure $\geq$90 mmHg, or use of antihypertensive drugs. Individuals with history or clinical evidence of angina, myocardial infarction, congestive heart failure, peripheral vascular disease, inflammatory disease, or any disease predisposing to vasculitis were excluded. Causes of secondary hypertension were excluded by appropriate investigations. Patients with stage 4 and 5 chronic kidney disease (GFR <30 mL/min/1.73 m$^2$) were also excluded. Medical history, including cardiovascular risk factors, previous and present cardiovascular events, and current medication regimen, was obtained during a personal interview and from medical files. Weight, height, and waist circumference were measured, and body mass index (BMI) was calculated. Brachial blood pressure was measured by a physician with a mercury sphygmomanometer after patients sat for 15 minutes or longer. The average of 3 measurements was used for the analysis.

## Blood and urine measurements

Venous blood samples were collected from all patients after 8 hours of overnight fasting for measurement. The blood samples were centrifuged at 3,000 rpm for 10 minutes immediately after collection, and the plasma samples were kept frozen at −70˚C until analysis. Each standard and plasma sample was analyzed twice, and the mean value was used in all subsequent analyses. The plasma high-sensitivity C-reactive protein (hs-CRP) level was determined using a latex-enhanced immunonephelometric assay (Dade Behring, Marburg, Germany). Plasma N-terminal pro b-type natriuretic peptide (NT-proBNP) was determined by a sandwich immunoassay (EIMA) with 2 antibodies (Cortez Diagnostics, Calabasas, CA). Plasma ADMA levels were measured by ADMA Fast Enzyme-Linked Immunosorbent Assay (ELISA) kit (DLD Diagnostika GMBH, Hamburg, Germany). Overnight urine samples were obtained for measurement of the albumin excretion rate. Normoalbuminuria was defined as albumin excretion rate of less than 20 mg/min, moderately increased albuminuria (previously known as microalbuminuria) was defined as albumin excretion rate of 20–200 mg/min, and severely increased albuminuria (previously known as macroalbuminuria) was defined as albumin excretion rate more than 200 mg/min.

## Endothelial microparticle extraction and measurement of NOX5

CD144+ microparticles were isolated as described with modifications [107, 108]. Briefly, Dynabeads G (Invitrogen, Carlsbad, CA) were washed with PBS containing 0.1% BSA and then reconstituted with PBS. Anti-CD144 antibody (Santa Cruz Biotechnology, Dallas, TX), which specifically targets endothelial cells, was mixed with prewashed Dynabeads G for 2 hours and then incubated with plasma samples at 1:200 dilution overnight at 4˚C. After precipitation, Dynabeads G were washed with PBS and 1% Tween-20 three times. The purity of

CD144+ MPs, determined by FACS analysis, was 70% ± 5.6%. With the use of FITC-conjugated beads as size references, the size of such particles was assessed to be <0.5 μm in diameter. Human NOX5 levels were measured using commercially available ELISA (Cusabio Technology LLC, Houston, Texas) kit according to the manufacturer's instructions. Samples were stored at −70˚C from the date of collection in 2008 until testing for NOX5 in 2014 (a total of 50 samples were available). The intra-assay and inter-assay variation coefficients of the tests were <8% and <10%, respectively. NOX1, NOX2, and NOX4 were also measured by ELISA kits (Cloud-Clone, Katy, TX) in another cohort of patients. The baseline characteristics of those patients are listed in S1 Table.

## Animals

Mice naturally do not express the NOX5 gene, therefore we have generated and validated humanized *Nox5* KI mice as previously described [37]. Briefly, the model was developed using the hypoxanthine phospho-ribosyl-transferase (Hprt) targeted transgenic approach under the control of the Tie2 promoter. Therefore, our *Nox5* KI mice express the NOX5 in endothelial and white blood cells which mimic the physiological human expression of NOX5. The expression of *Nox5* in the KI mice tissues was previously validated by real-time qPCR and compared to WT mice [37].

Age- and gender-matched groups of male and female mice (9–15 weeks old, $n$ = 19–20) and (68–87 weeks old, $n$ = 31–33) were used. All mice were allowed free access to water and food in a temperature-regulated room (22˚C) and placed in a 12-hour light-dark cycle.

## Blood pressure recording (telemetry)

*Nox5* KI and WT mice were anesthetized with isoflurane (induction, 3%–4%; maintenance, 1.5%–2.5%) and echocardiography (ultrasound) was performed (S11 Fig). To implant the telemetry transmitters, 5 days after the ultrasound, mice were anesthetized with the same protocol, and preoperative analgesia was done by subcutaneous injection of 0.05 mg/kg buprenorphine repeated every 12 hours. Each mouse was placed on a heating pad (UNO temperature control unit, UNO Roestvaststaal BV), and body temperature was monitored using a rectal probe and maintained at 37.0˚C using a feedback-controlled infrared light. An incision in the skin overlying the carotid artery was made. Via this incision, in the subcutaneous space of the flank a pocket was created for inserting the telemetry transmitter (TA11PA-C10; Data Sciences, St. Paul, MN) to monitor blood pressure, heart rate, and motor activity. The left carotid artery was dissected free and 3 ligatures (5–0, silk) were placed: at the bifurcation of the internal and external carotid to close the vessel, at the heart to temporarily close the vessel, and one in between to fixate the catheter. Via a small hole cut in the artery, the catheter was introduced and advanced into the aortic arch. Then, the pocket in the flank was filled with 3 mL prewarmed saline, and the transmitter was placed in the pocket. The wound was then closed using a polysorb 5–0 suture. All surgical procedures were performed under aseptic conditions. Post-operative analgesia was done by a subcutaneous injection of 0.05 mg/kg buprenorphine after 6 hours and 5 mg/kg carprofen after 24 and 48 hours. Mice were allowed to recover for 7–14 days before starting the measurement. Mice were housed individually in a quiet room. Blood pressure was measured over a 72-hour period [109–111], with 10 cycles of 75 s/hour. Radio signals from the transmitter were continuously monitored with a fully automated data-acquisition system (Dataquest A.R.T.; Data Sciences). Mice were euthanized by $CO_2/O_2$ inhalation, and organs were taken out for further analysis. Organ and body weights are presented in S3 Fig.

## Myograph

After mice were euthanized, thoracic aortae and femoral and saphenous arteries were dissected free from perivascular adipose tissue and mounted in a wire myograph (DMT, Aarhus, DK). The organ chamber was filled with Krebs-Ringer bicarbonate-buffered salt solution (KRB) that was continuously aerated with 95% $O_2$/5% $CO_2$ and maintained at 37˚C. The passive stretch procedure was performed to mimic the physiologically relevant internal lumen diameter as previously described [112]. Arterial contractile and relaxing responses were recorded at lumen diameters corresponding to a distending pressure of 100 mmHg in the thoracic aorta and femoral artery and at 90% of this diameter in the resistance-sized saphenous artery. This is justified because diastolic arterial blood pressure did not differ significantly between the aged KI and WT mice. In view of the comparable diameter–tension relationships, these diameters did not differ significantly between the 2 mouse strains (S12 Fig). The diameter tension relationships were constructed according to the law of Laplace for a cylindrical tube: $P = T/R$ (with P for transmural pressure, T for wall tension, and R for the lumen radius of the tube). Consequently, wall stiffness can be determined equally by recording (1) changes in tension in response to imposed changes in radius (wire myography) or (2) changes in radius in response to changes in transmural pressure (pressure myography) [113, 114]. Here, we used the former approach because it has a higher throughput than the latter. Stress-strain relationships and even better "incremental elastic (Young's) modulus" (which require additional recording of wall thickness) only come into play to discriminate between structural and material property contributions to changes in arterial stiffness. Part of the isolated thoracic aorta was studied in the absence and part in the continuous presence of 10 μM indomethacin to inhibit the production of prostaglandins that can act as endothelium-derived vasoactive factors in this vessel. The vessels were tested for their contractile response to 40 mM $K^+$, concentration-response curves of phenylephrine (0.01 to 100 μM) and endothelin-1 (1–256 nM) followed by Ach (0.01 to 100 μM), PAPA/NO (0.01–10 μM), or Bay60-2770 (0.01–10 μM)-induced relaxations. The wall tension of the vessel segment was continuously recorded with LabChart Pro (ADInstruments, Oxford, UK).

## Measurement of superoxide formation: DHE staining

Superoxide was measured in femoral arteries using the fluorescence dye DHE (Thermo Scientific Technology, the Netherlands). Frozen femoral arteries cryosections were fixated in 4% paraformaldehyde (PFA) in PBS and then incubated with 2 μM DHE for 30 minutes at 37˚C. After 3 washing steps with PBS, slices were incubated with DAPI (Sigma-Aldrich, the Netherlands) 2 μg/ml for 10 minutes. Sections were washed in PBS and then mounted using a Dako Fluorescence Mounting Medium (S3023, Agilent Technologies). Immunofluorescent signals were viewed using a Leica DMI3000 B fluorescence microscope. Some arteries were pretreated with 500 μM L-NAME for 30 minute at 37˚C before performing the DHE staining.

## RNA extraction, cDNA synthesis, and real-time qPCR

Thoracic aortae, femoral arteries, and saphenous arteries were isolated from mice and immediately submerged in RNAlater solution (Thermo Fisher Scientific). RNA was extracted using RNeasy Micro Kit (Qiagen) according to the manufacturer's protocol. cDNA was synthesized from 1 μg total RNA in 20 μl reactions using High Capacity cDNA Reverse Transcription Kit (Thermo Fisher Scientific). After synthesis, the cDNA was stored at −20˚C.

Real-time qPCR was performed on CFX96 Real-Time PCR Detection System (Bio-Rad). All reactions were performed in triplicates in a total volume of 20 μl each using TaqMan Universal PCR Master Mix (Applied Biosystems-Life Technologies) according to manufacturer

instructions. Three microliters of cDNA was used as a template, and pre-designed TaqMan primers of *β-actin* and *Nox5* were used. The specific assay IDs for the primers used are shown in S2 Table. The standard PCR conditions were as follows: 10 minutes at 95˚C, followed by 15 seconds at 95˚C and 1 minute at 60˚C, 59 repeats. The amount of mRNA was normalized to the measured expression of β-actin mRNA.

### Statistical analysis

All human and animal data are expressed as mean ± SEM for numeric variables and as number (percentage) for categorical variables. Comparisons of continuous variables between the 2 mice groups were performed by unpaired two-tailed Student *t* test and among the 3 human groups by one-way ANOVA followed by Tukey's multiple comparisons test (post hoc test). Comparisons of categorical variables among the human groups were assessed by a $\chi^2$ test. Comparisons between the 2 mice groups in telemetry data were done by two-way repeated measures ANOVA and in myograph by ordinary two-way ANOVA, followed by Sidak's multiple comparisons test. For the subgroup analysis of the NOX5 levels in human subjects, a frequency analysis was carried out, where the bin width is calculated using Sturges' rule [115]. To assess the modality of the data, the output frequencies were fitted with a single Gaussian and sum of 2 Gaussian distributions, as well as a two-tailed F-test with a null hypothesis as Gaussian and alternative hypothesis as a sum of 2 Gaussians was performed. Additionally, the adjusted $r^2$ values were compared to select the best-fitting distribution for the sample. Given the bimodal nature of the sample, the area under each Gaussian distribution was calculated using the formula "Amplitude*SD/0.3989," and subsequently the proportion of the NOX5 mechanotype was reported as the ratio between the 2 distributions. Data were analyzed using GraphPad Prism version 8.2 (GraphPad Software, San Diego, CA); $p < 0.05$ was considered to indicate statistical significance after multiple testing correction.

## Supporting information

**S1 Fig. Levels of NOX1, NOX2, NOX4, and ADMA in normotensive and hypertensive individuals.** (A–C) There were no differences in NOX1, NOX2, or NOX4 levels between hypertensive patients with normoalbuminuria ($n = 20$) compared to normotensive individuals ($n = 20$). NOX1 levels were higher (A), but NOX2 were lower (B) in hypertensive patients with microalbuminuria ($n = 20$) compared to normotensive individuals. Comparison between groups was done by one-way ANOVA followed by Tukey's multiple comparisons test. (D) ADMA levels were significantly higher in hypertensive patients ($n = 40$) compared to healthy individuals ($n = 20$). Comparison between the two groups was done by two-tailed unpaired *t* test. All data are represented as mean ± SEM, $*p < 0.05$, $***p < 0.001$. All raw data are included in the S1 Data file.
(TIF)

**S2 Fig. MAP in young and aged WT and KI mice.** (A) There was no significant difference in MAP between young WT ($n = 19$) and KI ($n = 20$). (B) Aged KI mice ($n = 33$) had higher MAP compared to WT ($n = 31$). Telemetry data were analyzed by two-way repeated measures ANOVA followed by Sidak's multiple comparisons test. All data are represented as mean ± SEM of *n* individual animals $**p < 0.01$. All raw data are included in the S1 Data file.
(TIF)

**S3 Fig. Body and organ weights in aged WT and KI mice.** (A–E) There was no difference in body, heart, and kidney weights between WT ($n = 24$) and KI mice ($n = 20$); however, lung/body weight ratio was higher in KI mice. Comparison between groups were done by two-tailed

unpaired $t$ test. All data are represented as mean ± SEM of $n$ individual animals $^*p < 0.05$.
(TIF)

**S4 Fig. Arterial stiffness in aged WT and KI mice.** (A–C) The relation between resting wall tension and arterial lumen diameter did not differ between KI ($n = 9$) and WT ($n = 9$) mice in thoracic aortae (A), femoral arteries (B), and saphenous arteries (C). All data are represented as mean ± SEM of $n$ individual animals. All raw data are included in the S1 Data file.
(TIF)

**S5 Fig. Ach-induced relaxations in arteries of aged WT and KI mice.** (A–C) Ach-induced relaxations were impaired in femoral arteries (A) of aged KI mice ($n = 9$) compared to WT ($n = 8$–9) but not in saphenous arteries (B) and thoracic aortae (with/without indomethacin) (C) precontracted with $K^+$. (D–F) Ach-induced relaxations were impaired in femoral arteries (D) of aged KI mice ($n = 9$) compared to WT ($n = 8$–9) but not in saphenous arteries (E) and thoracic aortae with/without indomethacin (F) precontracted with endothelin-1. (G) There was no difference in Ach-induced relaxations in thoracic aortae (with/without indomethacin) precontracted with phenylephrine between WT ($n = 9$) and KI mice ($n = 9$). (H–J) Ach-induced relaxations in arteries made to contract with endothelin-1 were reversed by 100 μM L-NAME in femoral arteries (H) and thoracic aortae (I), but not saphenous arteries (J) of both aged KI mice ($n = 8$–9) and WT ($n = 8$–9). Myograph data were analyzed by two-way ANOVA followed by Sidak's multiple comparisons test. All data are represented as mean ± SEM of $n$ individual animals, $^*p < 0.05$. All raw data are included in the S1 Data file.
(TIF)

**S6 Fig. qPCR of NOX5 in TAO, FAs, and SAs of aged KI mice.** There was no difference in NOX5 gene expression between the 3 vessel types ($n = 3$, each in duplicates). Comparison were done by one-way ANOVA. All data are represented as mean ± SEM of $n$ individual animals. All raw data are included in the S1 Data file. FA, femoral artery; SA, saphenous artery; TAO, thoracic aortae.
(TIF)

**S7 Fig. Contractile responses in arteries of aged WT and KI mice.** There was no difference in contractile responses to $K^+$, phenylephrine, and endothelin-1 in femoral arteries (A–C), saphenous arteries (D–F), and thoracic aortae (with/without indomethacin) (G–I) between WT ($n = 8$–9) and KI mice ($n = 9$). Comparison between 2 groups in contractile responses to $K^+$ was done by two-tailed $t$ test. Other myograph data were analyzed by two-way ANOVA followed by Sidak's multiple comparisons test. All data are represented as mean ± SEM of $n$ individual animals. All raw data are included in the S1 Data file.
(TIF)

**S8 Fig. Endothelium-independent relaxations in arteries of aged WT and KI mice.** (A–C) Relaxations induced by the NO donor, PAPA NO (0.01–10 μM), in femoral arteries (A), saphenous arteries (B), and thoracic aortae (with/without indomethacin) (C) did not differ between WT ($n = 8$–9) and KI mice ($n = 9$). (D) Relaxations induced by the apo-sGC activator, Bay60-2770 (0.01–10 μM), in femoral arteries did not differ between WT ($n = 4$) and KI mice ($n = 4$). Myograph data were analyzed by two-way ANOVA followed by Sidak's multiple comparisons test. All data are represented as mean ± SEM of $n$ individual animals. All raw data are included in the S1 Data file.
(TIF)

**S9 Fig. Representative images of DHE staining of femoral arteries of aged WT and KI mice.**
(TIF)

**S10 Fig. Ach-induced relaxations in femoral arteries of aged KI mice treated with antioxidants.** In segments of femoral artery ($n$ = 3–6) made to contract with 10 μM phenylephrine, relaxing effects of Ach (10 μM) were not reversed by 10 μM N-acetylcysteine (NAC) (A) or 100 μM tempol (B). Comparison between groups were done by two-tailed unpaired $t$ test. All data are represented as mean ± SEM of $n$ individual animals. All raw data are included in the S1 Data file.
(TIF)

**S11 Fig. Echocardiography in aged WT and KI mice.** There were no differences in all parameters between WT ($n$ = 28) and KI mice ($n$ = 29). Comparison between groups were done by two-tailed unpaired $t$ test. All data are represented as mean ± SEM of $n$ individual animals. All raw data are included in the S1 Data file.
(TIF)

**S12 Fig. Arteries diameter in aged WT and KI mice.** There were no differences in diameter of thoracic aortae (A), femoral arteries (B), and saphenous arteries (with/without indomethacin) between WT ($n$ = 9) and KI mice ($n$ = 9). Comparison between groups were done by two-tailed unpaired $t$ test. All data are represented as mean ± SEM of $n$ individual animals. All raw data are included in the S1 Data file.
(TIF)

**S1 Table. Baseline characteristics in healthy individuals and hypertensive patients of the second cohort.** Values are mean ± SD or number (%). BMI, body mass index; LDL, low density lipoprotein.
(XLSX)

**S2 Table. qPCR assays.**
(XLSX)

**S1 Data. Numerical raw data.** All numerical raw data are combined in a single Excel file, "S1_Data.xlsx." This file consists of several spreadsheets. Each spreadsheet contains the raw data of one table, figure, or subfigure as indicated.
(XLSX)

## Acknowledgments

We thank Daniel Marbach and Sven Bergmann for helpful discussions.

## Author Contributions

**Conceptualization:** Mahmoud H. Elbatreek, Emre Guney, Jan Baumbach, Jo G. R. De Mey, Harald H. H. W. Schmidt.

**Data curation:** Mahmoud H. Elbatreek, Sepideh Sadegh, Tim Kacprowski, Ahmed A. Hassan.

**Formal analysis:** Mahmoud H. Elbatreek, Sepideh Sadegh, Elisa Anastasi, Tim Kacprowski, Ahmed A. Hassan, Po-Hsun Huang, Chien-Yi Hsu, Pamela W. M. Kleikers.

**Funding acquisition:** Jan Baumbach, Harald H. H. W. Schmidt.

**Investigation:** Anil Wipat, Harald H. H. W. Schmidt.

**Methodology:** Mahmoud H. Elbatreek, Sepideh Sadegh, Elisa Anastasi, Emre Guney, Cristian Nogales, Tim Kacprowski, Ahmed A. Hassan, Po-Hsun Huang, Chien-Yi Hsu, Paul M. H. Schiffers, Ger M. Janssen, Pamela W. M. Kleikers.

**Project administration:** Mahmoud H. Elbatreek, Harald H. H. W. Schmidt.

**Resources:** Po-Hsun Huang, Chien-Yi Hsu, Paul M. H. Schiffers, Ger M. Janssen, Jo G. R. De Mey, Harald H. H. W. Schmidt.

**Software:** Sepideh Sadegh, Elisa Anastasi, Emre Guney, Cristian Nogales, Tim Kacprowski, Ahmed A. Hassan, Anil Wipat.

**Supervision:** Jan Baumbach, Jo G. R. De Mey, Harald H. H. W. Schmidt.

**Validation:** Po-Hsun Huang, Chien-Yi Hsu.

**Visualization:** Mahmoud H. Elbatreek, Cristian Nogales, Harald H. H. W. Schmidt.

**Writing – original draft:** Mahmoud H. Elbatreek, Jo G. R. De Mey, Harald H. H. W. Schmidt.

**Writing – review & editing:** Mahmoud H. Elbatreek, Emre Guney, Tim Kacprowski, Ahmed A. Hassan, Andreas Teubner, Jan Baumbach, Jo G. R. De Mey, Harald H. H. W. Schmidt.

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
