## [Editor Report · Decision Letter 0]

4 Feb 2020

Dear Dr Elbatreek, 

Thank you very much for submitting your manuscript entitled "NOX5-induced uncoupling of endothelial NO synthase is a causal mechanism and theragnostic target of an age-related hypertension endotype" for consideration as a Research Article by PLOS Biology. 

We were interested in the data showing that that Nox5-knock in mice had elevated blood pressure with age, with evidence for defective vasorelaxation in femoral arteries. Regrettably, however, given the known role of Nox5 in other models related to hypertension, including stroke and myocardial infarction, the study was not felt to represent the strength of advance that we seek in PLOS Biology papers. 

While we cannot consider your manuscript further for publication in PLOS Biology, we suggest, as an alternative, that you consider transferring this manuscript to PLOS ONE (http://journals.plos.org/plosone/). 

PLOS ONE is a peer-reviewed journal that accepts scientifically sound primary research. The review process at PLOS ONE differs from other PLOS journals in that it does not judge the perceived impact of the work or whether this falls within a particular area of research. Rather, it focuses on whether the study has been performed and reported to high scientific and ethical standards, and whether the data support the conclusions. This approach helps to eliminate the rejection cycles that authors commonly encounter when submitting to one journal after another. Please note that the journals are editorially independent and we therefore cannot guarantee the outcome if you choose to pursue a transfer.

If you would like to submit your work to PLOS ONE, please click the following link:

<DeepLinkData><DeepLinkTypeID>27</DeepLinkTypeID><peopleID>1214449</peopleID><userSecurityID>6e2a942e-1360-4ac1-bf03-ac9748786e60</userSecurityID><documentID>39237</documentID><revision>0</revision><manuscriptNumber>PBIOLOGY-D-20-00252</manuscriptNumber><docSecurityID>3bca3841-0fd2-4088-bd08-6386fb3369fd</docSecurityID></DeepLinkData>

If you do NOT wish to submit your work to PLOS ONE, please click this link to decline: 

<DeepLinkData><DeepLinkTypeID>28</DeepLinkTypeID><peopleID>1214449</peopleID><userSecurityID>6e2a942e-1360-4ac1-bf03-ac9748786e60</userSecurityID><documentID>39237</documentID><revision>0</revision><manuscriptNumber>PBIOLOGY-D-20-00252</manuscriptNumber><docSecurityID>3bca3841-0fd2-4088-bd08-6386fb3369fd</docSecurityID></DeepLinkData>

Should you choose to transfer your submission to PLOS ONE, you will receive a confirmation email within 24-48 hours of accepting the transfer. Please note, all PLOS journals are editorially independent and vary in submission requirements. Your submission details and manuscript files will be transferred automatically; once in the PLOS ONE Editorial Manager site, your submission will be returned to you and you will be asked to provide additional information before you can finalize your new submission to PLOS ONE. If you have any questions, please feel free to contact the journal at plosone@plos.org.

Thank you for giving us the opportunity to consider your work.

Sincerely,

Di Jiang,

Associate Editor

PLOS Biology

---

## [Decision Letter · Decision Letter 1]

20 Mar 2020

Dear Dr Elbatreek,

Thank you very much for submitting your manuscript "NOX5-induced uncoupling of endothelial NO synthase is a causal mechanism and theragnostic target of an age-related hypertension endotype" for consideration as a Research Article at PLOS Biology. Your manuscript has been evaluated by the PLOS Biology editors, an Academic Editor with relevant expertise, and by four independent reviewers.

In light of the reviews (below), we will not be able to accept the current version of the manuscript, but we would welcome re-submission of a much-revised version that takes into account the reviewers' comments. We cannot make any decision about publication until we have seen the revised manuscript and your response to the reviewers' comments. Your revised manuscript is also likely to be sent for further evaluation by the reviewers, and we might involve new reviewers, should the original reviewers become unavailable, to assess your response to the comments in the first round of review.

We expect to receive your revised manuscript within 2 months. 

**IMPORTANT - SUBMITTING YOUR REVISION**

*Re-submission Checklist*

*Published Peer Review*

*PLOS Data Policy*

*Blot and Gel Data Policy*

Sincerely,

Di Jiang, PhD

Associate Editor

PLOS Biology

REVIEWS:

Reviewer #1: In this study, mice with Nox5 knocked in were studied and found to have hypertension with age. Some evidence for NOS uncoupling was shown. There are multiple questions that arise upon reading the manuscript which detract. 

Your statement in the intro regarding renal artery stenosis as the only cause of hypertension with a known mechanism is wrong. Other known causes include adrenal adenomas, pheochromocytomas and numerous single gene mutations involving renal transporters. 

The statement: " . . . antihypertensive therapy is neither curative nor effective, needing high numbers to treat with many patients still experiencing adverse outcomes" doesn't make sense. Do you mean that therapy requires high doses of drugs or several agents to achieve target pressures?

The machine learning/prunned network analysis is only hypothesis generating. An important experimental finding is that Nox5 is highly expressed in human microparticles. It is essential that the investigators show that there is not an increase in other Nox isoforms in these microparticles. 

It seems odd that there is no cardiac hypertrophy in the Nox5 KI mice given that they are hypertensive. 

It is unclear how the diameter tension relationships were constructed in figure S2. It seems these are just tensions at different degrees of stretch in the wire myograph. This is a very crude way to determine stress strain relationships, which should be analyzed in pressurized vascular segments as described by Humphrey and co-workers.1 It is very unusual that there is not vascular stiffening in the Nox5 mice, if they are really hypertensive. 

The evidence that sepiapterin improves endothelium-dependent vasodilatation is only partial support that NOS uncoupling has occurred. Can the authors show an increase in vascular superoxide production that is inhibited by L-NAME? Can they demonstrate that the NOS3 enzyme migrates as a monomer on low temperature gel? Would treatment with sepiapterin lower blood pressure?

The differences in endothelium-dependent vasodilatation between the femoral artery (affected) and the saphenous artery and aorta (not affected) is hard to understand. It is known that branch vessels are less dependent on NO, but this should affect the saphenous and femoral arteries similarly. 

The degree of acetylcholine mediated relaxation in the saphenous artery (figure 3b) seems very impaired. Normally vessels of this size relax at least by 60 to 75%. This is also true for the responses in 3A. 

What is the significance of the inhibition of PE contraction by indomethacin? This would suggest a role of some vasoconstrictor prostanoid, but which one? How does this relate to Nox5? 

1. Ferruzzi J, Bersi MR, Humphrey JD. Biomechanical phenotyping of central arteries in health and disease: advantages of and methods for murine models. Ann Biomed Eng. 2013;41:1311-1330.

Reviewer #2: In this manuscript, the authors show that hypertensive, microalbuminuric human subjects have increased expression of NOX5. Mouse NOX5KI exhibit increases in systolic BP with aging, and treatment of vasculature from these mice vasodilate in response to sepiapterin. The authors conclude that it is NOX5 that is mediating the hypertenison in both humans and mice. 

Suggestions for revision:

1. The authors should show telemetry BP in mice for at least 5 days and show mean arterial BP (MAP). Does MAP change with aging in NOX5KI mice compared to controls. 

2. Does treatment of mice with sepiapterin decrease BP in aging NOX5KI mice?

These studies are necessary to support the authors' hypothesis.

Reviewer #3: The paper entitled NOX5-induced uncoupling of endothelial NO synthase is a causal mechanism and theragnostic target of an age-related hypertension endotype by Elbatreek et al describes a novel role of the superoxide-generating NOX5 isoform as a cause of increased blood pressure in aged mice overexpressing the human NOX5 isoform (NOX5 KI mice)and finds a correlation of increased NOX5 levels in endothelial microparticles with hypertensive patients as well as a potential mechanism by an in silico interactome approach showing a close interation between NOX5 and nitic oxide signalling using human data. Finally, using an ex-vivo pharmacological approach using vessels purified from NOX5 KI mice, they confirm the connection between NOX5-derived ROS and the uncoupling of endothelial NO synthase by recoupling NOS with sepiapterin

The paper is well-written and the methods are sound. It uses human and mouse in vivo and ex vivo data and provides several novel important findings, in particular, the measurement of circulating microparticular NOX5 as a biomarker in a subpopulation of hypertensive patients, providing valuable tool for stratification of potential future clinical trials. Also the spontaneous hypertensive phenotype in NOX5 KI mice is of great interest for further testing of preclinical drugs. Finally the mechanism described is of great interest-at least in this model.

Altogether, the paper can be recommended for publication if the comments below are answered satisfactorily.

Major: 

1. Several statements in the discussion are very strong and mix the findings from human clinical samples and results obtained with mice overexpressing NOX5, a rather artificial system. For example, the mechanism of uncoupling NO synthase by NOX5 has indeed not be proven in humans. The discussion would benefit from a more balanced interpretation and should also highlight potential pitfalls of the animal model.

2. In a previous paper (Casas et al, 2019) describing the NOX5 KI model, the same authors show high expression levels of NOX5 in white blood cells. First, this fact should be documented in the present study and the authors should comment on how they exclude a potential effect of leukocytic NOX5 on hypertension in aged animals as the ex vivo data do not solely exclude this.

3. Addition of antioxidants or the pharmacological tool DPI to the ex vivo experiments would be an asset. Has it be done? Have these compounds been tested? These data should be added whatever the outcome and interpreted accordingly.

4. NOX5 is known to be expressed in leucocytes. Is it possible that some contaminating white blood cells are present in endothelial microparticles. How was this excluded?

Minor:

Page 12, figure S4. Page 13, lanes 216-225: document in the text the concentrations used for PAPA/NO, BAY 60-2770 and sepiapterin

Page 17, lane 297: Here NOX5 is important….: sentence is not clear. What the "here" refer to?

Page 15-17. What are the chances of administering sepiapterin in patients?

Page 22. It is nowadays standard practice to document the sex of the animals used. Have both genders been used in the study? Was blood pressure similar in males and females? Was there a reason to select one gender over another one?

Reviewer #4: Interactome modularity based analysis is acceptable.

Figure is not of sufficiently good resolution. The printout was not readable. Please provide high resolution image.

---

## [Decision Letter · Decision Letter 2]

10 Aug 2020

Dear Dr Elbatreek,

Thank you for submitting your revised Research Article entitled "NOX5-induced uncoupling of endothelial NO synthase is a causal mechanism and theragnostic target of an age-related hypertension endotype" for publication in PLOS Biology. I have now obtained advice from the original reviewers and have discussed their comments with the Academic Editor. Please note that reviewer 4 was replaced by a new reviewer 5, who was granted access to the peer-review history.

Based on the reviews, we will probably accept this manuscript for publication, assuming that you will modify the manuscript to address the remaining points raised by the reviewers. Please also make sure to address the data and other policy-related requests noted at the end of this email.

We expect to receive your revised manuscript within two weeks. Your revisions should address the specific points made by each reviewer. Please submit the following files along with your revised manuscript:

In addition to the remaining revisions and before we will be able to formally accept your manuscript and consider it "in press", we also need to ensure that your article conforms to our guidelines. A member of our team will be in touch shortly with a set of requests. As we can't proceed until these requirements are met, your swift response will help prevent delays to publication.

*Copyediting*

*Published Peer Review History*

*Early Version*

*Submitting Your Revision*

Sincerely,

Gabriel Gasque

Senior Editor

on behalf of

Ines Alvarez-Garcia, PhD,

Senior Editor,

ialvarez-garcia@plos.org,

PLOS Biology

ETHICS STATEMENT:

Please indicate within your manuscript if your human experiments approved by the research ethics committee of Taipei Veterans General Hospital adhered to the Declaration of Helsinki or any other specific national or international ethical guidelines. 

Please include the full name of the IACUC/ethics committee that reviewed and approved the animal care and use protocol/permit/project license. Please also include an approval number.

Please include the specific national or international regulations/guidelines to which your animal care and use protocol adhered. Please note that institutional or accreditation organization guidelines (such as AAALAC) do not meet this requirement.

DATA POLICY:

Please update your S1 Data File to provide the quantitative summary data for Fig 1CD.

Please ensure that each figure legend in your manuscript include information on where the underlying data can be found, and ensure your supplemental data file/s has a legend.

Reviewer remarks:

Reviewer #1: The authors have responded to prior concerns. There continue to be a few statements that are incorrect that could be easily modified.

It is not true that Nox1 has not been shown to have a role in pre-clinical studies of hypertension. Dikalova et al showed that overexpression of this isoform potentiated hypertension in mice (PMID 16230485). Gawazzi et al showed that Nox1 deficient mice had a complete loss of the sustained hypertensive response to angiotensin II. 

The authors should note that the hypertensive response with aging in the Nox5 ki mice is very similar to that observed in mice overexpressing p22phox in the vascular smooth muscle and in mice lacking the vascular smooth muscle ecSOD (see PMID 26595812). 

These issues should be briefly discussed. 

Reviewer #2: In this manuscript the authors provide evaluation of NOX5 and NOS3 based on human interactome analyses. They go on to develop a mouse NOX5 gene knock in into endothelial and WBCs. They find that the aging NOX5 KI mice are hypertensive, but not the young adult animals. They claim that it is the muscular conduit vessels that mediate the hypertension in the aging KI. The manuscript takes many liberties with the analyses of previous studies that are not quite correct. They also interpret data to show the importance of conduit vessels (lines 198-99) without ever measuring endothelial function in resistance vessels, particularly in the kidney that are known to control sodium handling and hypertension. Furthermore, they do not see antioxidants, tempol or N-acetyl cysteine, having any effect on vascular function/hypertension in the KI mice. The lack of antioxidant protection against the hypertension suggests that there is no NO being produced likely due to age-related endothelial dysfunction to cause a response when the ROS are removed. The authors discuss the role played by H2O2 as a vasodilator, without mentioning peroxynitrite which is a much better vasodilator.

Other issues:

1. "eNOS KO mice do not have cardiac hypertrophy". Whole body eNOS KO is developmentally lethal, isn't it?

2. "no difference in BP between males and females". The data should be shown since there are only 9 mice of both sexes that were used for MAP. 

3. In the human studies, why were so few women included? 

4. The authors mention that NOX1 and NOX2 do not cause hypertension unless prohypertensive drugs are given (lines 65-67)--antioxidants reduce BP in SHR and Dahl salt sensitive rats, that do not have NOX5.

5. The figures as shown in the manusscript are not of publication quality.

Reviewer #3: The authors replied to the comments satisfactorily

Reviewer #5: This paper investigates the causal link between NOX5 gene in humans and age-related pulmonary hypertension via uncoupling of endothelial NO synthase. The study starts with computational prediction of PPI networks, proceeds with analysis of control and PH patients and ends with mouse knock-in experiments (mice do not have endogenous NOX5 gene). I found the mouse results convincing, although I cannot say the same for the human results for the following reasons.

The human IID database cannot map three of the seed proteins (GUCYA1, GUCYA2, GUCYB1) and reports no experimentally validated PPIs for NOX3. So it is strange that the identified "NOX5 subnetwork; Fig 1) includes both GUCYA1 and GUCYB1. Is it a nomenclature issue? If so, the correct gene names need to be reported in the paper. If not, an explanation is warranted. Is it that they did not look for networks with experimental support? If so, they should correct the manuscript to reflect the actual search they performed.

Furthermore, NOX5 had only one study to support its PPIs. The authors should discuss the limitations of their predictive model that is based on such limited evidence. Subsequently any claims about "causality" (in the sense of direct causal effects) should be toned down as they may constitute an overstatement of the results.

Fig 1 is badly presented and it is unclear if it supports the authors' claims. First the resolution makes the axes labels unreadable. We can only guess what the authors want to say. Second, Fig 1C and 1D should probably be reversed, because one first identifies the bimodal distribution among hypertension patients and then compares the groups (unless the groups were identified with external characteristics, like albuminuria). Third, the lines with the asterisks in fig. 1C do not extend to make clear what they compare these groups to. Are they compared to the controls? Fourth the actual p-values should be provided after correction. From the bi-modal distribution presentation in Fig. 1D it looks like the first group of hypertension patients may not be so different than controls.

I wish I could clearly read Fig. 1 which is very important and especially fig 1D. What is the Y-axis? The labels are unreadable and the legend/text does not explain it clearly either. And this is supposed to be a very important evidence because they use the bimodal distribution to split the hypertensive patients into two groups. If what they measure is circulating NOX5 from endothelial cells then the first group may be not significantly different from the normal.

The control patients include no smokers and no diabetics. Smoking actually represents a significant difference between the two cohorts, but it is not discussed at all. It is well know that smoking impairs NO synthase and especially in endothelial cells. Eg:

https://www.ahajournals.org/doi/full/10.1161/01.atv.16.4.546

https://www.atsjournals.org/doi/full/10.1165/ajrcmb.19.5.3091

Beta-blockers and a number of other molecules are differing significantly between the three groups and should be discussed as more direct causes (or results) of the observed differences.

So overall I would say this is a very good mouse study but the importance of these results in humans is something that requires further investigation.

---

## [Editor Report · Decision Letter 3]

18 Sep 2020

***EDIT THIS LETTER BEFORE SENDING***

Dear Dr Elbatreek,

On behalf of my colleagues and the Academic Editor, Cecilia W Lo, I am pleased to inform you that we will be delighted to publish your Research Article in PLOS Biology. 

Early Version

PRESS 

Kind regards,

Vita Usova

Publication Assistant, 

PLOS Biology

on behalf of

Ines Alvarez-Garcia,

Senior Editor

PLOS Biology